# Lipid Dys-Homeostasis Contributes to APOE4-Associated AD Pathology

**DOI:** 10.3390/cells11223616

**Published:** 2022-11-15

**Authors:** Adina-Nicoleta Lazar, Linda Hanbouch, Lydie Boussicaut, Baptiste Fourmaux, Patricia Daira, Mark J. Millan, Nathalie Bernoud-Hubac, Marie-Claude Potier

**Affiliations:** 1Univ Lyon, INSA Lyon, CNRS, LaMCoS, UMR5259, 69621 Villeurbanne, France; 2ICM Paris Brain Institute, CNRS UMR7225, INSERM U1127, Sorbonne University, Hôpital de la Pitié-Salpêtrière, 47 Bd de l’Hôpital, 75013 Paris, France; 3Institut De Recherche Servier IDRS, Neuroscience Inflammation Thérapeutic Area, 125 Chemin de Ronde, 78290 Croissy-sur-Seine, France; 4Institute of Neuroscience and Psychology, College of Medical, Vet and life Sciences, Glasgow University, 68 Hillhead Street, Glasgow G12 8QB, Scotland, UK

**Keywords:** lipid homeostasis, APOE4, Alzheimer’s disease, Aβ peptide, tau

## Abstract

The association of the APOE4 (vs. APOE3) isoform with an increased risk of Alzheimer’s disease (AD) is unequivocal, but the underlying mechanisms remain incompletely elucidated. A prevailing hypothesis incriminates the impaired ability of APOE4 to clear neurotoxic amyloid-β peptides (Aβ) from the brain as the main mechanism linking the apolipoprotein isoform to disease etiology. The APOE protein mediates lipid transport both within the brain and from the brain to the periphery, suggesting that lipids may be potential co-factors in APOE4-associated physiopathology. The present study reveals several changes in the pathways of lipid homeostasis in the brains of mice expressing the human APOE4 vs. APOE3 isoform. Carriers of APOE4 had altered cholesterol turnover, an imbalance in the ratio of specific classes of phospholipids, lower levels of phosphatidylethanolamines bearing polyunsaturated fatty acids and an overall elevation in levels of monounsaturated fatty acids. These modifications in lipid homeostasis were related to increased production of Aβ peptides as well as augmented levels of tau and phosphorylated tau in primary neuronal cultures. This suite of APOE4-associated anomalies in lipid homeostasis and neurotoxic protein levels may be related to the accrued risk for AD in APOE4 carriers and provides novel insights into potential strategies for therapeutic intervention.

## 1. Introduction

Alzheimer’s disease (AD) is the most common cause of dementia and a major public health concern, with >130 million cases worldwide anticipated by 2050. Late-onset sporadic AD (sAD) represents more than 90% of AD cases, and is associated with age-related cellular and molecular alterations in the brain. Two main neuropathological features characterize AD: the accumulation of aggregates of amyloid β (Aβ) peptides in extracellular brain parenchyma and in the perivascular areas (senile plaques and amyloid angiopathy, respectively) and intracellular fibrils of hyper-phosphorylated tau (the neurofibrillary tangles) [1]. In addition to these two defining hallmarks, numerous studies indicate an implication of lipid metabolism in the development and the progression of the disease. The presence of abnormal lipid granules in neuroglia was noticed from the time of Alois Alzheimer [2,3,4]. Later on, lipidomic approaches revealed alterations in the level of several specific lipids in AD brains, from cholesterol and ceramides to plasmogens [5,6,7]. Genome-Wide Association Studies (GWAS) and Transcriptome-Wide Association Studies (TWAS) [8,9] later identified several lipid-related genes associated with sAD pathology. Among them, APOE, low-density lipoprotein receptors LRP1, acyltransferases ACAT1, ABCA1 and ABCA7 transporters, and the enzyme CYP46A1 [10,11,12,13,14,15,16] may be highlighted. In fact, *APOE ε4* allele—encoding the lipid binding apolipoproteinE—is the strongest genetic risk factor for AD [17], and *APOEε4* carriers are more susceptible to brain amyloid burden, even at asymptomatic stages [18]. The APOE4 isoform is less efficient in clearing Aβ from the brain to periphery [19,20,21], enhancing the conversion of the peptide into toxic species and increasing plaque load in brain parenchyma [22,23]. The APOE4 isoform was also associated with an exacerbation of tau pathology in a mouse model of taupathy [24] and altered blood–brain barrier integrity [25]. Furthermore, astrocytes and neurons carrying the *APOEε4* allele have decreased lipid-binding capacity [26]. Taken together, these findings point to a cumulative effect of APOE genotype combined with altered lipid homeostasis in AD pathology.

Prevailing evidence points, thus, to a strong link between *APOE* genotype and lipid metabolism in the progression of AD but this relationship is not yet fully understood. Studies on human *APOE3/4* knock-in (KI) mice revealed no effect of the genotype on brain levels of cholesterol and intermediates/metabolites (lathosterol, desmosterol and oxidized metabolites) [27,28]. The global level of phospholipids in the brain of APOE3 and APOE4-KI mice was examined in a separate study that found no major effect of the genotype on the abundance of different classes of phospholipids [29]. Another study focused on the analysis of fatty acids in the hippocampus and cortex of male and female APOE3 and APOE4-KI mice. The analysis revealed an age-related decrease in the levels of omega-3 fatty acid only in APOE4 females, but no alterations in males, underlining an interplay between sex, age and genotype [30]. All these investigations have not, then, proven sufficient to elucidate the cellular substrates of the interplay between APOE genotype, lipid homeostasis and AD pathology.

In order to further identify new links between *APOE* genotype and lipid metabolism and to evaluate their co-joint impact on AD pathology, we first characterized the brain lipidomic profile of several classes of lipids in KI mice expressing human APOE3 or APOE4. We focused on three major and most relevant classes of membrane lipids: phosphatidylcholines, phosphatidylethanolamines and cholesterol. We also investigated the global composition in fatty acids, the essential components of most lipids, in order to embrace all lipid classes in the brain including the less abundant triglycerides and cholesteryl esters. We further examined the link between lipid dysregulations and the levels of related transcripts based on a recently published RNAseq data set [31].

To evaluate whether differences in lipid composition between the APOE4 and APOE3 genotypes were associated with AD pathology, we measured the levels of Aβ, tau and hyper-phosphorylated tau (p-tau) in primary neurons.

## 2. Materials and Methods

### 2.1. Sample Preparation and Lipid Extraction

Half forbrain tissues from adult B6.129P2-*Apoe^tm3(APOE*4)Mae^* N8 and B6.129P2-Apoetm3(APOE34)Mae N8 12 month-old male mice were collected and snap frozen at −80 °C, homogenized by cryogenic grinding of frozen samples, using a mortar and pestle cooled up with liquid nitrogen. Mice were ordered from TACONIC (www.taconic.com).

Total brain homogenates (approx. 130 mg) were extracted according to the Folch procedure (FOLCH et al., 1957) [32]: 2 mL of cold methanol-BHT (butyl-hydroxytoluene) spiked with several standards (50 µg or PC di-17:0, PE di-17:0 and, 5 µg of cholesterol-2,3,4-^13^C) was added to the homogenate and stirred for 10 min, at 4 °C. The homogenates were further incubated with 4 mL of chloroform for 2 h at 4 °C and the mixture was adjusted with 1.5 mL of water. It was further vortexed for 30 s and centrifuged for 5 min at 2000 rpm. The lower (organic) phase was removed and a second extraction was performed by adding 4 mL of chloroform to the remaining aqueous phase. The two extracts were combined and dried under nitrogen (protocol adapted from Sharman et al., 2010 [29]). Lipid extracts resolubilized in chloroform/methanol 2:1 (*v*/*v*) were dispatched in three loads for targeted lipidomics (cholesterol and derivatives, phospholipids and fatty acids).

### 2.2. Lipid Analysis

Lipid analysis was performed on the Functional Lipidomics platform acknowledged by IBiSA (Infrastructure in Biology, Health and Agronomy).

#### 2.2.1. Analysis of Cholesterol and Derivatives by GC-MS/MS

Cholesterol was isolated from the total lipid extract by thin layer chromatography as follows: 50 µL of total lipid extract in chloroform was loaded on a silica plate, along with several standards: cholesterol, triglycerides, phospholipids, diglycerides and cholesteryl esters. A mixture of hexan:diethylether:glacial acetic acid (80:20:1, *v*/*v*/*v*) was used as eluent. The different classes of lipids were revealed by spraying the plate with 0.02% dichlorofluorescein in ethanol. The band corresponding to free cholesterol was retrieved by scratching with a cutter and cholesterol and its derivatives were extracted from the silica gel using chloroform. The dry residue was derivatized with BSTFA (*N, O-bis (trimethylsilyl)trifluoroacetamide*) and then analyzed by gas chromatography (HP 7890B, Agilent Technologies, Santa Clara, CA, USA) coupled with triple quad mass spectrometry (GC-MS/MS) using the electron impact ionization (EI) mode (7000 C, Agilent Technologies, Santa Clara, CA, USA).

GC-MS/MS was equipped with a SolGel-1ms fused silica capillary column (Trajan, SGE; 60 m × 0.25 mm). The carrier gas was helium (1.2 mL/min). The temperature of the split/splitless injector was set at 280 °C. The oven temperature program was as follows: 55 °C for 4 min, followed by 40 °C/min up to 250 °C, and 20 °C/min up to 310 °C (for 25 min). The samples were injected in a splitless mode. The temperatures of the mass spectrometer transfer line and the source were set at 250 and 20 °C, respectively. Nitrogen was used as collision gas (1.5 mL/min). The electron ionization energy was 70 eV. Cholesterol was detected using the MRM (multiple-reaction monitoring) mode. Peak detection, integration and quantitative analyses were executed using MassHunter software (Agilent Technologies, Santa Clara, CA, USA).

#### 2.2.2. Phospholipids Analysis by Electrospray Mass Spectrometry (ESI-MS)

The phospholipids (PL) were isolated from the total lipid extract, in parallel to cholesterol, by thin layer chromatography. The band corresponding to PL was retrieved by scratching and the lipids were extracted from the silica gel using toluene-methanol (60:40). They were dried under N_2_ atmosphere and resolubilized in methanol for LC-MS analysis.

High-performance liquid chromatography was performed using a Shimadzu Nexera LC-30 equipped with an autosampler, a binary pump and a column oven. The analytical column was a WATERS CORTECS HILIC (1.6 μm, 3.0 × 150 mm). Separation was carried out at 40 °C using a linear gradient. Mobile phase A consisted of 50% 10 mM ammonium formate and 50% acetonitrile. Mobile phase B was 5% 10 mM ammonium formate and 95% acetonitrile. The mobile phase gradient was delivered as follows: 100% B from 0 to 2 min, 0% B at 10 min then return to the initial conditions. The flow rate was 0.7 mL/min and 5μL sample volume was injected. The HPLC system was coupled on-line to a QTRAP 4500 (Sciex) equipped with electrospray ionization source (ESI). Source parameters were as follows: source temperature was set at 350 °C, curtain gas at 40, gas 1 at 20 and gas 2 at 35, using nitrogen. Analyses were achieved in the positive mode, based on precursor ion scan (*m*/*z* 184) for PC and neutral loss (*m*/*z* 141) for PE. Spray voltage was set at 4500 V. Nitrogen is used as collision gas. Finally, peak detection, integration and quantitative analyses were performed using Analyst and LipidView softwares (Sciex, Framingham, MA, USA).

#### 2.2.3. Analysis of FA Composition by GC

A volume of 50 µL of total lipid extract was evaporated and resolubilized in 500 µL toluen-methanol (1:1, *v*/*v*) for methylation. The methylation reaction was performed in the presence of 500 µL BF_3_/methanol (14%) used as catalyzer, under N_2_ atmosphere, by heating at 100 °C for 90 min. The reaction was stopped by fast cooling of the samples and the addition of K_2_CO_3_ (10% in water). Isooctane was further added to extract the methylated fatty acids—the upper organic phase; the extraction was performed 2 times. The samples were dried under N_2_ and re-solubilized in isooctane for GC analysis.

GC was performed on an HP 6890 (Agilent Technologies) instrument equipped with a fused silica capillary BPX70 column (Trajan, SGE; 60 m × 0.25 mm). The carrier gas was hydrogen (1 mL/min). The temperatures of the flame ionization detector and the split/splitless injector were set at 250 and 230 °C, respectively. The oven temperature program was as follows: 50 °C for 2 min, followed by 20 °C/min up to 140 °C, and 2 °C/min up to 240 °C (for 5 min). The samples were injected in a splitless mode. Peak detection, integration and quantitative analyses were executed using MassHunter software (Agilent Technologies, Santa Clara, CA, USA).

### 2.3. Cultures of Primary Neurons

Primary cortical neurons cultures were performed by dissecting P0-P1 B6.129P2-*Apoe^tm3(APOE*4)Mae^* N8 and B6.129P2-*Apoe^tm3(APOE34)Mae^* N8 new born mice as described before (Boussicault et al., 2018) [33]. Cortices were incubated in L15 medium (Invitrogen) containing 0.05% trypsin/EDTA (Gibco, Thermo Fisher Scientific, Waltham, MA, USA) for 8 min at 37 °C. Mechanical dissociation was applied to the cells kept in Neurobasal (without glutamine, Gibco) with 2% B27 (Gibco), 1% N2-supplement (Thermofisher scientific, Waltham, MA, USA), 1% Glutamax (Gibco) and 1% penicillin/streptomycin (Gibco, and DNAse I (10,000 U/mL, Serlabo, Entraigues-sur-la-Sorgue, France) and 30% FCS (Fetal Calf Serum). Cells were centrifuged 5 min at 152× *g* at 4 °C, plated in 6 well microtiter plates coated with poly-D-lysine (0.1 mg/mL, Sigma Aldrich, St. Louis, MO, USA) and grown in 2% B27, 2 mM glutamax, 1% penicillin/streptomycin (all from Life Technologies) at 37 °C 5% CO_2_. Half of the medium was changed at DIV7. After 5, 9 and 12 days in vitro (DIV5, DIV9 and DIV12), supernatants were aliquoted in 1.5 mL polypropylene tubes (Corning, Corning, NY, USA) containing a protease inhibitor cocktail (Roche, Penzberg, Germany) and were then stored at −80 °C.

### 2.4. Aβ 38, 40 and 42 Measurements

Concentrations of the Aβ38, Aβ40 and Aβ42 species of β-amyloid peptide were measured by multiplex Electro-Chemiluminescence Immuno-Assay (ECLIA). Assays were performed according to the manufacturer’s instructions. Briefly, samples were analyzed using Meso Scale Discovery (MSD) SECTOR™ Imager 2400 (Meso Scale Discovery, Gaithersburg, MD, USA), with the Rodent Aβ triplex kit (also from MSD); carbon 96-well plates contained four capture spots in each well, one of which was blocked with BSA (as standard curve control), and the others were coated with isoform-specific anti-Aβ antibodies specific for Aβ38, Aβ40, Aβ42, respectively. A volume of 100 μL of blocking buffer solution was added to all wells to avoid non-specific binding. The plates were then sealed, wrapped in tin foil, and incubated at room temperature on a plate shaker (600 rpm) for 1 h. Wells were then washed three times with washing buffer, and 25 μL of the standards (Aβ38, Aβ40, and Aβ42) and samples were then added to the wells, followed by an Aβ-detecting antibody at 1 μg/mL (MSD) labelled with a Ruthenium (II) trisbipyridine N-hydroxysuccinimide ester; this detection antibody was 4G8 (which recognizes the epitope Aβ18-22 of the human and rodent peptide). Plates were then aspirated and washed 3 times. The MSD read buffer (containing TPA) was added to wells before reading on the Sector Imager. A small electric current passed through a microelectrode present in each well, producing a redox reaction of the Ru2+ cation, emitting 620 nm red light. The concentration of each Aβ isoform was calculated for each sample, using dose–response curves, with the blank being cell-less culture medium.

### 2.5. Western Blotting

Primary neurons were washed with 1X PBS and then lysed on ice using RIPA buffer (50 mM Tris-HCl pH 8.0, 150 mM sodium chloride, 1.0% NP-40, 0.5% sodium deoxycholate, and 0.1% sodium dodecyl sulfate) (Sigma Aldrich), to which were added phenylmethylsulfonyl fluoride 100X (Sigma Aldrich) and a cocktail of inhibitors of proteases (Complete Mini, Roche). Lysates were sonicated 3 times for 5 min then stored at −80 °C. Protein concentration of the lysates was quantified by the Bradford assay (Biorad) according to the manufacturer’s instructions. Western blots were made from the cell lysates. Proteins from cultured lysates were separated in 16.5% Tris-Tricine (Biorad) polyacrylamide gels. Proteins were transferred to a polyvinylidene difluoride membrane (Biorad) at 150 V for three hours at 4 °C. After 1 h of saturation with 10% milk, membranes were incubated overnight at 4 °C with primary antibodies directed against the Amyloid Precursor Protein APP (APPCter-C17 [34]), actin β (Sigma Aldrich), total tau (B19 [35]) diluted 1/2000 and Phospho-tau (AT8 Thermofisher) diluted 1/1000. Membranes were incubated with fluorescent goat secondary antibodies (diluted to 1/10,000 in 0.05% TBS-tween solution) anti-mouse (LI-COR) for anti-actin and AT8 and anti-rabbit (LI-COR) for anti-APP, total tau and APOE, for 1 h under agitation at room temperature and away from direct light. The revelation and quantification of fluorescence were carried out by Odyssey’s analysis software (Set up ImageStudio CLx) (Odyssey Clx LI-COR).

### 2.6. Data Processing

Cholesterol and derivatives content in the brain were calculated per g of tissue with respect to 2,3,4-^13^C cholesterol used as internal standard. Average and standard error were estimated over 7 samples for the APOE3 genotype and 8 samples for the APOE4 genotype (one APOE3 sample was formerly used for the preliminary setup). For the rest of the analysis, 8 samples per phenotype were used. The proportion of PC and PE was calculated as the molar percentage of the total phospholipid content (PC + PE). Only species present in at least 14 of the 16 samples were analyzed. The relative amount of the excluded species was below 1% of the global amount of phospholipids and should not have affected the outcome of analyses. Relative contents of individual species within a class of phospholipids were estimated as the molar percentage of total lipid of the respective class. Only the species with a relative abundance greater than 1% for at least one of the samples were considered. The proportion of different fatty acids was also expressed as the molar percentage of total fatty acids detected. Details on lipidomic analysis are available in (Appendix A). Statistical analysis of lipids was performed with Origin Pro software (Origin Lab), using multiple paired student test and one-way analysis of variance (ANOVA) followed by multiple comparison tests with Ficher’s correction. Statistical significance was defined as: * *p <* 0.05, ** *p* < 0.005 and *** *p* <0.001.

## 3. Results

### 3.1. APOE Genotype Disturbs Cholesterol Turnover

The main role of APOE in the brain is to transport cholesterol. To determine if there is a direct link between APOE genotype and cholesterol production and transport, we investigated the level of cholesterol in the brain of human APOE3 and APOE4-KI mice. The abundance of free cholesterol in the brain of APOE4 or APOE3 mice was similar (Figure 1A), as previously described in the literature [36]. Still, in primary cortical neurons derived from APOE4 mice, the level of cholesterol was significantly lower compared to APOE3 neurons, suggesting altered cholesterol homeostasis.

We also inspected the level of cholesterol precursors and derivatives in the brains of the two isogenic mice. In addition to cholesterol, we detected two of its derivatives: desmosterol, a precursor of cholesterol, and *beta*-cholestanol, a metabolite of cholesterol. Significantly lower levels of desmosterol and β-cholestanol were found in APOE4 mice compared to APOE3 mice (Figure 1A). These findings suggested impaired cholesterol homeostasis associated with *ε4* allele. In the brain, cholesterol level is regulated by synthesis, metabolism (degradation) and transport within a cell and from cell to cell. Thus, in order to evaluate cholesterol turnover, we measured the ratios cholesterol/desmosterol and cholesterol/β-cholestanol (Figure 1B). The ratios cholesterol/desmosterol and cholesterol/β-cholestanol were significantly increased in APOE4 mice compared to APOE3 mice, suggesting an imbalance between cholesterol and the other sterols.

### 3.2. Levels of PC and PE and Changes in Their Composition in APOE4-KI Mice

APOE is not only a transporter of cholesterol but also of phospholipids in the brain. We investigated the impact of *APOE* genotype on the main classes of phospholipids in the brain: phosphatidylcholines (PC) and phosphatidylethanolamines (PE). Both classes showed a similar profile in the brain of APOE3 mice (Figure 2A). In APOE4 mice, the proportion of PC is considerably increased with respect to PE, the ratio PC/PE being significantly higher with respect to APOE3 mice (Figure 2B). We further investigated potential alterations in the composition of PC and PE for the two *APOE* genotypes. The most abundant molecular species of PC is PC (16:0/18:1) followed by PC (16:0/16:0) and PC (18:0/18:1). In APOE4 mice, PC (16:0/18:1) was significantly depleted (Figure 2C). Still, the level of the other phosphatidylcholine species bearing long-chain fatty acids or the degree of unsaturation was similar for APOE3 and APOE4 mice.

*APOE* genotype had an opposite effect on PE (16:0/18:1) that was significantly upregulated in APOE4 mice while PE species bearing the 22:6 fatty acid—an important omega-3 fatty acid in the brain—were significantly depleted (Figure 2D).

In addition, significantly higher levels of PE species bearing monounsaturated fatty acids (MUFAs) and lower levels of PE species with polyunsaturated fatty acids (PUFAs) were observed, particularly omega-3 fatty acids. Actually, information concerning changes in the degree of unsaturation of PE fatty acids in *APOE4* allele is limited although it is clear that PUFAs are critical for normal neuronal functioning and neuroprotection [37,38,39].

### 3.3. APOE Genotype and Fatty Acid Profile and Metabolism

In order to determine whether *APOE* genotype alters the global balance of fatty acids in brain lipids or only in phospholipids, we have profiled their extent in APOE3 and APOE4-KI mice. The level of three fatty acids was dependent on APOE genotype: palmitic acid (16:0) was significantly decreased in APOE4 mice while oleic acid (18:1n-9) and nevronic acid (24:1n-9) were enriched (Figure 3A). This enrichment was associated with a significant increase in the level of MUFAs, especially omega-9 fatty acids, counterbalanced by a decrease (although non-significant) in PUFAs and saturated fatty acids (Figure 3B,C). There was no interaction between *APOE* genotype and the global level of omega-3 and omega-6 fatty acids. These unsaturated fatty acids are important components of cell membrane and involved in complex metabolic pathway.

### 3.4. Lipid Dysregulations and Gene Expression in APOE Genotypes

The expression of genes involved in the metabolism of the dysregulated lipids was analyzed using a recently published RNAseq data set obtained on the same mouse models [31] (Appendix A). Several of the genes involved in cholesterol synthesis and metabolism showed a reduced expression in APOE4 mice: DHCR7—the reductase involved in the final steps of cholesterol synthesis, HSD3b—that converts cholesterol to *beta*-cholestanol, and CYP46A1—that controls cholesterol removal from the brain (Figure 4). The expression level of different genes involved in phospholipid production and processing was also altered in the APOE4 genotypes. The Pcyt2 gene, belonging to the cytidylyltransferase family, a key regulator of PE synthesis was under-expressed, while the Pcty1b gene, related to PC synthesis, was overexpressed in APOE4 mice males. Moreover, two of the genes involved in the rate control of FA metabolism from the family of elongases (ELOVL—ELongation of Very Long fatty acids) were also altered; the level of expression of ELOVL5 (known to have a certain preference for PUFAs) [40] was lower, while the one of ELOVL7 (particularly selective for saturated or MUFAs) [41] was significantly increased in APOE4 males.

### 3.5. Neurons of APOE4-KI Mice Produce a Higher Amount of Aβ Peptide

The effect of *APOE* genotype on the capacity of neurons to produce Aβ peptides was investigated on primary cortical neurons of APOE3 and APOE4-KI mice. We observed a significant enrichment in the three main Aβ fragments: 38, 40 and 42 in APOE4- as compared to APOE3-derived neurons (Figure 5). Moreover, the overaccumulation of the peptides in the culture media in APOE4 neurons was heightened overtime. Aβ38 was undetectable at div 5 (5 days in vitro).

We observed a significant enrichment of the three main Aβ fragments: 38, 40 and 42 in APOE4- as compared to APOE3-derived neurons that was not due to an overexpression of the Amyloid Precursor Protein (APP). Indeed, levels of APP protein measured by Western blotting were similar in APOE3- and APOE4-derived neurons (Figure 6A). We also investigated the level of intracellular APOE. APOE was significantly reduced in APOE4-derived neurons; this could favor Aβ accumulation due to impaired clearance by the lipoprotein [42] (Figure 6B).

We also questioned the involvement of APOE in tau pathology in our model. Levels of total tau were largely increased in APOE4- compared to APOE3-neurons. Similarly, levels of pathological phosphorylated tau were higher in APOE4-derived neurons (Figure 7). Thus, APOE4 was also able to heighten tau pathology as well.

## 4. Discussion

The *APOEε4* allele is the most prominent genetic risk factor for AD. Heterozygous carriers have a 3-fold higher AD risk, while homozygous individuals have a 15-fold higher AD risk [43], corresponding to a prevalence of 30% of AD cases at the age of 75 and over 50% by the age of 85 [44]. Many hypotheses have been proposed to explain the relationship between APOE4 and the pathogenesis of AD, most of them pointing to Aβ transport and clearance, which culminates with increased accumulation, spread and deposition in the brain [45,46]. Nonetheless, APOE may play a role in pathology not only downstream of Aβ production but also upstream by influencing APP processing. Since evidence for a connection between lipid homeostasis, APOE, Aβ and tau pathology remains sparse [36,47], in this study, we questioned to what extent APOE variants may influence brain lipid homeostasis, Aβ peptide production and tau hyper-phosphorylation. In contrast to previous lipidomic studies that focused only on a particular class of lipids, either cholesterol or fatty acids, here we present a quantitative assessment of the most abundant lipids classes in the brain: cholesterol, phosphatidylcholines and phosphatidylethanolamines. We also established the global fatty acid profile to get an overview of all lipid classes. We have, furthermore, correlated the observed lipid dysregulations with the level of expression of several genes involved in lipid metabolism in addition to the pathological markers of AD such as Aβ and tau/phospho-tau. One limitation of this study is that we focused only on male mice. Taking into account interactions between APOE genotype, lipid content and gender [30], it would be of interest to undertake further studies in females as well.

### 4.1. Altered Cholesterol Turnover in APOE4-KI Mice

We first evaluated the homeostasis of cholesterol in the brains of APOE4- compared to APOE3-KI mice. We mainly detected cholesterol and two other sterols (desmosterol and *beta*-cholestanol). Desmosterol is a precursor of cholesterol while *beta*-cholestanol is a metabolite of cholesterol degradation. Levels of lathosterol and 7-dehydrocholesterol, the alternative precursors of cholesterol, were probably too low to be detected in APOE KI mice. These data confirm that synthesis of cholesterol mainly uses the desmosterol pathway also known as the Bloch pathway (Figure 8) [48,49]. The amount of cholesterol in brains of APOE4 mice was slightly diminished with respect to isogenic APOE3 mice (although not statistically significant), with desmosterol and *beta*-cholestanol showing significantly lower levels. Therefore, the ratios between cholesterol/desmosterol and cholesterol/*beta*-cholestanol were higher in APOE4 mice compared to APOE3. The decreased levels of desmosterol in APOE4 brains may correspond to reduced de novo cholesterol synthesis, mirroring observations in AD brains [50,51]. Indeed, the gene encoding DHCR7, an enzyme involved in downstream cholesterol synthesis (Figure 8A), was under-expressed. On the other hand, lower levels of *beta*-cholestanol could be explained by restricted degradation of cholesterol in order to ensure its required efflux to neurons [52,53]. The reduced expression of the gene encoding HSD3β7 and CYP46A1 enzymes (both involved downstream to cholesterol synthesis, Figure 8A) suggests the necessity to maintain a certain level of cholesterol for transport to neurons, as previously established [54,55,56]. Indeed, neurons derived from APOE4-KI mice showed significantly lower levels of cholesterol with respect to APOE3 ones.

The present study suggests that cerebral cholesterol turnover is impaired in APOE4-KI mice. Moreover, the effect of the *APOEε4* allele on brain desmosterol and *beta*-cholestanol levels has not been reported previously.

### 4.2. APOE4 Genotype Is Associated to Lower Levels of Phosphatidylethanolamines Bearing Polyunsaturated Fatty Acids

We further investigated alterations in the composition of the two major phospholipids. We observed only modest modifications in the amount of PC and PE in APOE4 mice with respect to APOE3 mice, but a significant increase in the PC/PE ratio. These two phospholipids share similar but parallel steps in their biosynthesis involving two rate limiting enzymes (Figure 8B): phosphoethanolamine cytidylyltransferase (PECT)—encoded by Pcyt2 and phosphocholine cytidylyltransferase (PCCT)—encoded by Pcyt1b, to generate CDP-ethanolamine and CDP-choline intermediates [57]. A secondary pathway is the direct conversion of PE to PC via phosphatidylethanolamine-N-methyltransferase (PEMT). The enhancement of PC/PE ratio observed in APOE4 mice was related to the first two key enzymes, as transcripts of Pcyt2 and Pcyt1b showed altered expression in the APOE4 genotypes: Pcyt1b -encoding PCCT was upregulated while Pcyt2, encoding PECT was under-expressed. PEMT gene expression was similar for the two genotypes (Appendix A). In line with these observations, increased PCCT levels in AD brain (not linked to APOE phenotype) have previously been documented [58] as well as an elevated PC/PE ratio in AD mice, particularly in the hippocampus [59].

In addition to alterations of the ration of the two phospholipids, the APOE4 isoform was also associated with changes in the array of their fatty acids. The precise balance between saturation and unsaturation/polyunsaturation determines the biomechanical properties of membranes and, consequently, the handling of membrane-associated proteins. PC and particularly PE had altered levels of long-chain fatty acids (from 16 to 24 C atoms), either monounsaturated or polyunsaturated. MUFA-PE and the oleic acid-PE were elevated in APOE4. PUFA-PE and particularly PE containing omega-3 fatty acids were, on the contrary, significantly depleted. It is known that plasma saturated FA and MUFA levels increase with aging, while the level of PUFAs decreases [60,61]. Saturated fatty acids provide a certain rigidity to membrane and MUFAs have a low capacity to fluidify plasma membrane [62,63]. PUFAs, in reverse, and especially omega-3 FA, are able to fluidify plasma membranes and to diminish lipid raft clustering [64,65].

**Figure 8 cells-11-03616-f008:**
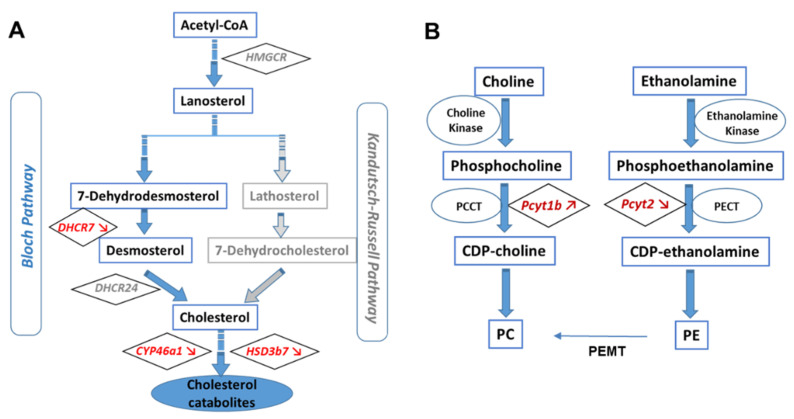
(**A**) Pathway of cholesterol synthesis in the brain from acetylCoA reduction by HMGC reductase, through the action of 7-dehydrocholesterol reductase (DHCR7) and 3β-hydroxysterol 24-reductase (DHCR24); its catabolism involves the formation of various catabolites by the actions of several enzymes such as cytochrome P450 family 46 subfamily A member 1 (CYP46a1) and hydroxy-delta-5-steroid dehydrogenase, 3 beta- and steroid delta-isomerase 7 (HSD3b7); (**B**) Pathway of PC and PE synthesis [66]. Abbreviations: PCCT—phosphocholine cytidylyltransferase, encoded by Pcyt1b; PECT—phosphoethanolamine cytidylyltransferase, encoded by Pcyt2; CDP-choline—cytidil-5-diphosphocholine; CDP-ethanolamine—cytidil-5-diphosphoethanolamine; PC—phosphatidylcholine; PE—phosphatirylethanolamine; PEMT—phosphatidylethanolamine *N*-methyltransfera.se.

### 4.3. Elevated Levels of MUFAs, Particularly Omega-9 FA, in APOE4-KI Mice

The global levels of several other fatty acids were also altered in APOE4-KI mice. We observed elevated levels of two MUFAs—particularly nevronic acid (24:1). A similar link between APOE4 and MUFA overload was previously described [67] as well as an increase in MUFAs with age [68]. Omega-9 FA and particularly nevronic acid have been previously associated with AD pathology [69]. Interestingly, levels of nevronic acid were significantly elevated in mid-frontal cortex, temporal cortex and hippocampus of AD patients [70]. We found that the APOE4 genotype was correlated with overexpression of ELOVL7 gene, encoding the enzymes involved in fatty acids elongation (particularly MUFAs), while ELOVL5—involved in the elongation of PUFAs—was downregulated [71]. These alterations in gene expression were consistent with the amount of available FA substrates (significant enrichment in MUFAs and reduction in PUFAs) [71]. Lipid metabolism involving synthesis, elongation, desaturation, oxidation and degradation of fatty acids is a complex processing involving a series of redundant enzymes with preferences for one particular lipid substrate. Thus, variations in the expression levels of these enzymes do not impact one lipid species in particular, but generate interconnected modifications of the abundance of several FA. The precise quantification of these interactions is technically challenging and will require additional study.

### 4.4. Interrelationship between Lipid Dys-Homeostasis and AD Pathological Markers

In order to link these dysregulations in lipid homeostasis with the pathophysiology of AD, we investigated the levels of different pathological markers in primary neurons of APOE3/APOE4-KI mice, as they are the main source of Aβ and tau production in the brain [72,73].

We observed a marked increase in Aβ levels in primary neurons derived from APOE4 brains as compared to APOE3. As the level of APP was unchanged, the increased content of Aβ peptides was apparently not caused by the overexpression of their substrate. The levels of tau and hyper-phosphorylated tau in neurons derived from APOE4-KI mice was also doubled with respect to the APOE3 genotype. Moreover, we observed lower levels of APOE in primary neurons of APOE4 mice, similar to findings in APOE3- and APOE4-KI mice brain reported by Mann et al. [36], suggesting lower capacity in Aβ clearance.

APOE4 is known to be less efficient in shuttling cholesterol to neurons than its isoform APOE3 [26]. In view of the lower level of cholesterol in the neurons, the rate of cholesterol shuttling via lipoproteins from the astrocytes could be expanded, as already demonstrated [74]. The augmented frequency in cholesterol loaded-lipoproteins uptake in APOE4 neurons may increase APP internalization and recycling, and as a consequence, Aβ production [75]. This possibility is supported by several studies showing that lipid-poor APOE can increased Aβ production by augmenting APP internalization and recycling [76,77]. Our observations also reinforce a previous report that a deficiency in cholesterol turnover correlates with increased tau phosphorylation and microtubule depolymerization in axons [78]. All together, these studies suggest that cholesterol dys-homeostasis in APOE4 genotype impacts AD pathological markers via its turnover and not by directly interfering with APP processing. Abnormally low content of PUFA-PE in APOE4 mice potentially leads to the formation of rigid lipid platforms [79,80] that favor the encounter between APP and its processing enzyme BACE and as a consequence Aβ production [81,82,83]. Lipid rafts can also promote accumulation of Aβ oligomers and of hyper-phosphorylated tau [84,85]. Moreover, lower levels of PUFAs, particularly omega-3 PUFAs, impair the counter-regulation of neuro-inflammation. Indeed, resolvins, protectins and other lipoxines are derived from PUFAs and are critical in modulating inflammation and oxidative stress [86], important for alleviating Aβ and tau induced toxicity [87]. Likewise, lower levels of PE with respect to PC, as observed in the APOE4 genotype may induce a dysfunction in autophagy, and thus compromise the ability to degrade and clear neuro-toxic proteins [86,87]. Finally, the dysregulations observed in the global composition of fatty acids aggravate the detrimental effects discussed above: that is, the increase in oleic acid load stimulates the activity of gamma-secretase (the second cleaving enzyme involved in APP processing), augmenting Aβ production in mice [88] and promotes pathological actions of tau, as observed in vitro [89].

## 5. Conclusions

In conclusion, this study reveals a diverse pattern of dysregulation of lipid homeostasis in APOE4 mice: notably, impaired cholesterol turnover, an imbalanced PC/PE ratio, lower levels of omega-3-bearing PE and disrupted levels of several fatty acids. These alterations were associated with changes in the expression of several genes linked to lipid metabolism. They were also related to increased levels of Aβ, tau and hyper-phosphorylated tau protein. In view of these observations, it is possible that readjusting lipid homeostasis may represent a novel direction for opposing and/or delaying the pathological events triggering AD. Future studies should examine more closely the precise molecular mechanisms interlinking lipid dysregulations with the APOE4 genotype, in particular the altered generation and processing of cholesterol and phospholipids. Pathways linking lipid dys-homeostasis to the anomalous generation of neurotoxic peptides and other pivotal cellular substrates of AD also merit further characterization.

## Figures and Tables

**Figure 1 cells-11-03616-f001:**
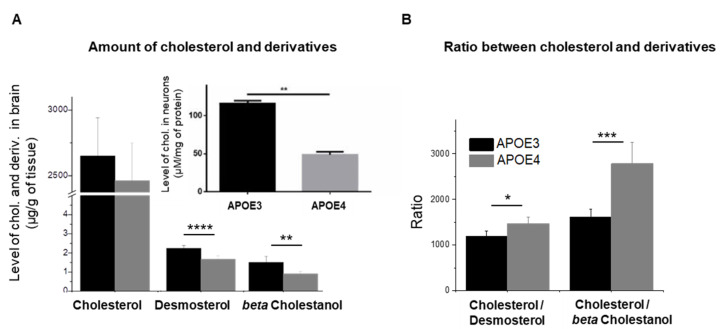
Level of cholesterol and other sterols in the brain of APOE3 and APOE4-KI mice: (**A**) amount of different sterols detected; the insert shows the level of intracellular cholesterol in primary neurons. (**B**) Ratio between cholesterol and derivatives; the results are means of seven or eight samples/APOE genotype. Statistics: Multiple paired *t*-test *: *p* < 0.05, **: *p* < 0.01, and ***: *p* < 0.001, ****: *p* < 0.0001.

**Figure 2 cells-11-03616-f002:**
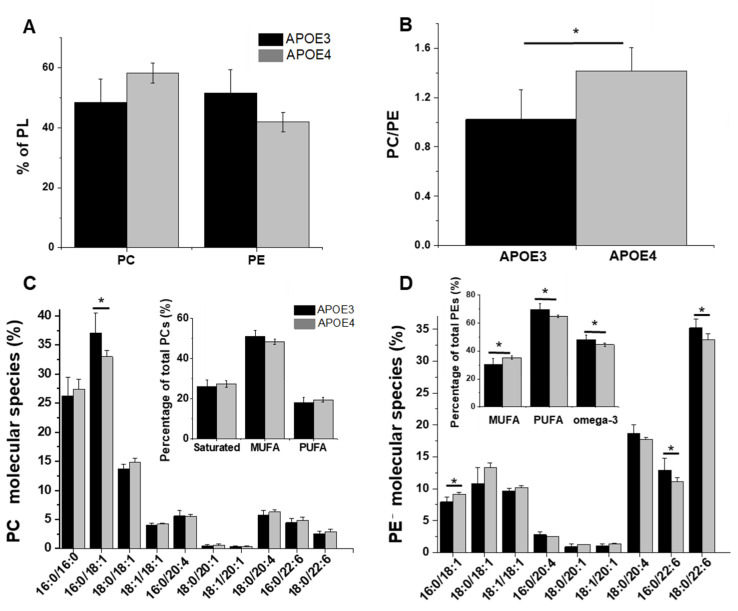
Distribution of the two major classes of phospholipids detected in APOE3 and APOE4 mice: (**A**) percentage of PC and PE in the brain of APOE3 and APOE4 mice; (**B**) the ratio PC/PE; (**C**) molecular species of PC and percentage of PC bearing saturated, monounsaturated (MUFAs) or polyunsaturated fatty acids (PUFAs); (**D**) molecular species of PE and percentage of PE bearing monounsaturated (MUFAs), polyunsaturated fatty acids (PUFAs) or omega-3 fatty acids. Statistics: one-way ANOVA with Ficher’s correction *: *p* <0.05.

**Figure 3 cells-11-03616-f003:**
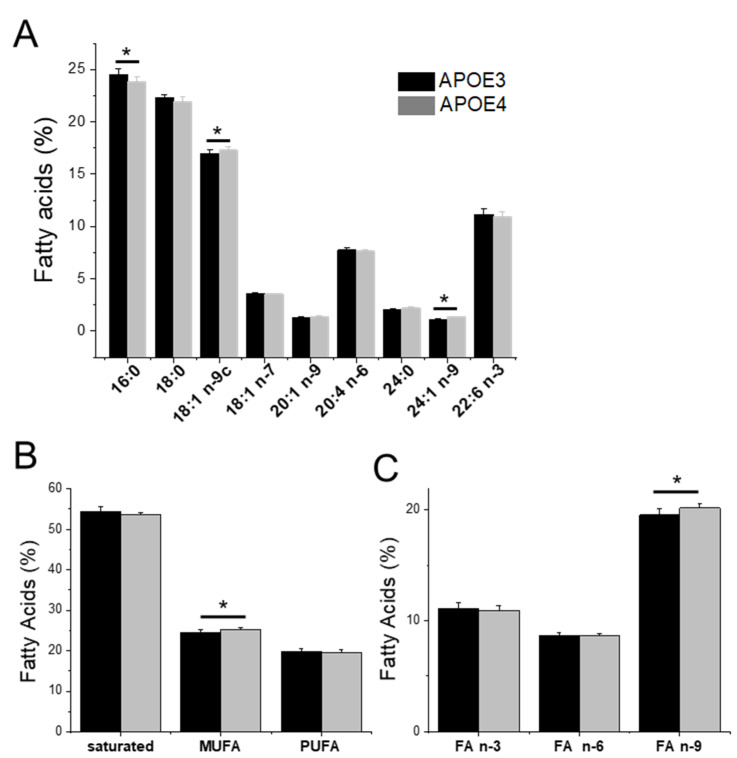
Fatty acids of the total lipid extract of APOE3 and APOE4 mice: (**A**) overview of the fatty acid composition; (**B**) level of fatty acids classified as a function of their degree of unsaturation; (**C**) proportion of unsaturated FA. Statistics: one-way ANOVA with Ficher’s correction *: *p* < 0.05.

**Figure 4 cells-11-03616-f004:**
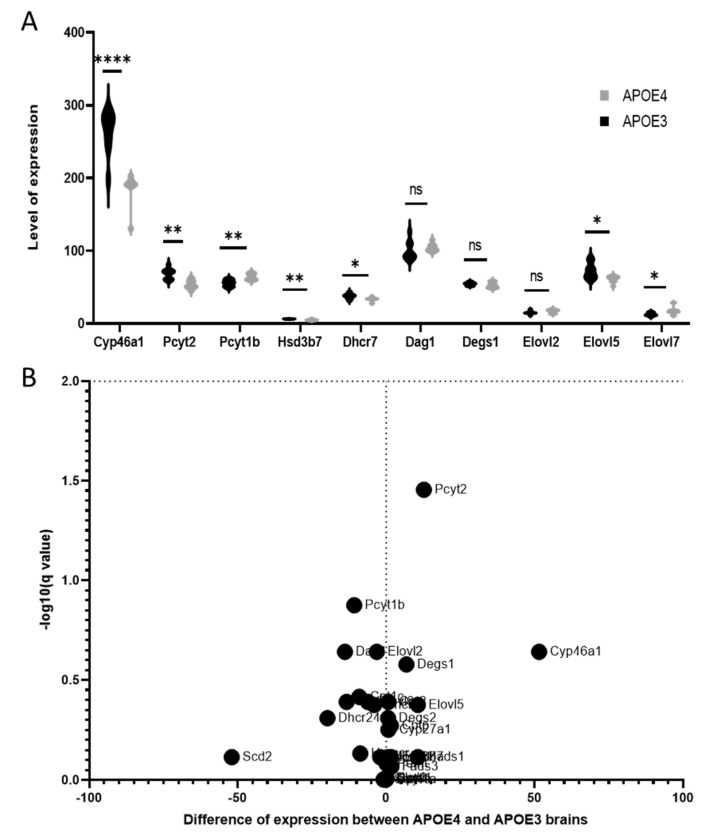
Gene expression of enzymes involved in lipids pathway in the brain of KI mice carrying the human APOE3 or APOE4 alleles [31]. (**A**) Level of expression analysis with multiple unpaired *t*-test *: *p* < 0.05, **: *p* < 0.01, and ****: *p* < 0.0001. (**B**) Volcano plots.

**Figure 5 cells-11-03616-f005:**
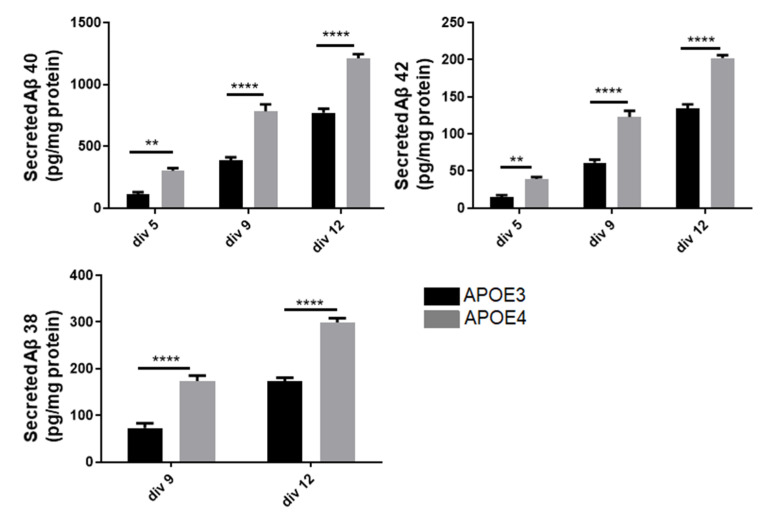
Effects of APOE genotype on Aβ level in primary neurons. Primary cortical neurons cultures were obtained from P0-P1 newborn KI mice carrying the human APOE3 or APOE4 alleles. After 5, 9 or 12 days in culture (DIV5, DIV9 and DIV12), levels of Aβ38, Aβ40 and Aβ42 were assessed in the culture medium using the MSD multiplex ELISA (pg/mg of proteins measured in cells lysates). Statistics: two-way ANOVA mean +/− SEM; **: *p* < 0.01, and ****: *p* < 0.0001, comparison with APOE3, 3 independent cultures with 2 to 3 replicates per culture.

**Figure 6 cells-11-03616-f006:**
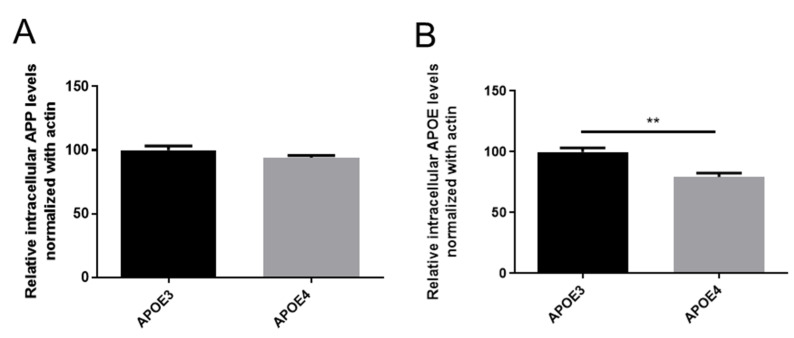
Effects of APOE genotype on intracellular levels of APP (**A**) and APOE (**B**). Primary cortical neurons cultures were obtained from P0-P1 newborn KI mice carrying the human APOE3 or APOE4 alleles. After 12 days in culture, APOE and APP levels were assessed in cell lysates using Western blot and normalization with actin levels. Statistics: Mann and Whitney test **: *p* < 0.01, comparison with APOE3, 3 independent cultures with 2 to 3 replicates per culture.

**Figure 7 cells-11-03616-f007:**
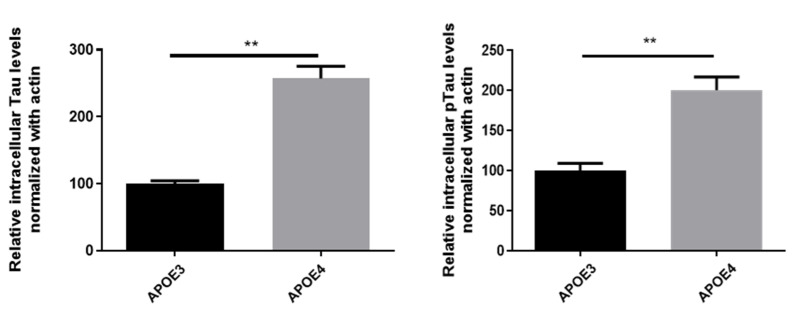
Effects of APOE genotype on intracellular levels of Tau and pTau. Primary cortical neurons cultures were obtained from P0-P1 newborn KI mice carrying the human APOE3 or APOE4 alleles. After 12 days in culture, tau and p-tau levels were assessed in cell lysates using Western blot and normalization with actin levels. Statistics: Mann and Whitney test **: *p* < 0.01, comparison with APOE3, 3 independent cultures with 2 to 3 replicates per culture.

## Data Availability

The data that support the findings of this study are available in Appendix A.

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
