# Peer review of "Lipid Dys-Homeostasis Contributes to APOE4-Associated AD Pathology"

_cells, 2022, doi:10.3390/cells11223616_

Round 1

Reviewer 1 Report

I congratulate the authors for a paper that is very well written interesting and well structured. I have a few minor comments/ and one concern:

11.      In the Abstract,  Page 1, line 27: I would prefer the use of the world “altered” rather than the used “deficient”. The APOE 4 genotype in humans have even been associated with certain cognitive advantages in youth, and could represent an example of antagonistic pleiotropy. (ref Evans et al. Neurobiology of aging 35 (2014) 1615-1623, for further refs).

22.  In the Introduction, Page 2 line 80-95: This section would be improved if it was shortened, not going into details of the methods used.  That info is already available in the methods section.

     Page 2-3, line 96-103: These lines are a summary of results- (well written), but in my opinion , do not belong in the Introduction section.

     3. In the Results

Page 7, line 320: ”APOE genotype had a strong effect ..”  I would use another word than ”strong”. Suggestion: ”… a significant ( p<0,05) effect”.

Page 9, section 3.4 Lipid dysreglations and gene expression in APOE genotypes. And fig 4.

I have some difficulties understanding this section.  I did not find any explanation in the Materials and Methods section.  Did you use the RNAdata from another study/ not the same mice as in your own?  Could this be made clearer?  However interesting and  important finding.

Author Response

I congratulate the authors for a paper that is very well written, interesting and well structured. I have a few minor comments/ and one concern:

We thank the reviewer for these positive comments.

  1.    In the Abstract, Page 1, line 27: I would prefer the use of the world “altered” rather than the used “deficient”. The APOE 4 genotype in humans have even been associated with certain cognitive advantages in youth, and could represent an example of antagonistic pleiotropy. (ref Evans et al. Neurobiology of aging 35 (2014) 1615-1623, for further refs).

We thank the reviewer for the remark (and the reference). We implemented the modification as suggested. 

  1. In the Introduction, Page 2 line 80-95: This section would be improved if it was shortened, not going into details of the methods used.  That info is already available in the methods section.

 As suggested by the  reviewer, we have curtailed the details on the methodology used for lipidomic analysis as follows (see line 82-94, page 2): In order to further identify new links between APOE genotype and lipid metabolism and to evaluate their co-joint impact on AD pathology, we first characterized the brain lipidomic profile of several classes of lipids in KI mice expressing human APOE3 or APOE4. We focused on three major and most relevant classes of membrane lipids: phoshatidylcholines, phosphatidylethanolamines and cholesterol. We also investigated the global composition in fatty acids, the essential components of most lipids, in order to embrace all lipid classes in the brain including the less abundant triglycerides and cholesteryl esters. We further examined the link between lipid dysregulation and the levels of related transcripts based on a recently published RNAseq data set [31].

To evaluate whether differences in lipid composition between APOE4 and APOE3 genotypes were associated with AD pathology, we measured the levels of Aβ, tau and hyper-phosphorylated tau (p-tau) in primary neurons.”

     Page 2-3, line 96-103: These lines are a summary of results- (well written), but in my opinion , do not belong in the Introduction section.

As suggested, we have removed this paragraph. Indeed, the conclusion section already provides an overview of the results.

  1. In the Results

Page 7, line 320: ”APOE genotype had a strong effect ..”  I would use another word than ”strong”. Suggestion: ”… a significant ( p<0,05) effect”.

We made the modification in the text (page 7, line 313) as follows: “APOE genotype had an opposite effect on PE (16:0/18:1) that was significantly upregulated in APOE4 mice…”

Page 9, section 3.4 Lipid dysreglations and gene expression in APOE genotypes. And fig 4.

I have some difficulties understanding this section.  I did not find any explanation in the Materials and Methods section.  Did you use the RNAdata from another study/ not the same mice as in your own?  Could this be made clearer?  However interesting and  important finding.

As stated in the introduction: “We further examined the link between lipid dysregulations and the levels of related transcripts based on a recently published RNAseq data set [31].” We also specify in the results section 3.4 that this data set was obtained on the same mouse models” (line 350-351, page 9).

Reviewer 2 Report

In the manuscript by Lazar et al., the authors measured lipid metabolism in the brain APOE3 and APOE4 mice via Functional Lipidomics platform. They found APOE influenced cholesterol turnover and phospholipid composition in an isoform dependent manner, whereas the expression levels of several lipid metabolism related genes also changed significantly. The results are interesting and informative. There are several comments about the current version of this manuscript.

1. The authors should clarify some of the limitations of this study in the discussion part, like only male mice were used, lack of APOE heterzygous mice results, and etc.  

2. The authors detected cholesterol and phospholipid metabolism changes in APOE4 mice. How about the APOE level in mouse brain? Is there any correlation between lipid levels and APOE levels?

3. Using primary neurons, the authors measured Abeta, p-Tau and apoE levels, it's not new that APOE4 increases Abeta and p-Tau levels in neurons.  The novelty of the manuscript will be enhanced if the authors could measure the lipid changes in primary neurons using lipidomics analysis and evaluate how much neuronal lipid contributes to the lipid changes of the whole brain. .

Author Response

In the manuscript by Lazar et al., the authors measured lipid metabolism in the brain APOE3 and APOE4 mice via Functional Lipidomics platform. They found APOE influenced cholesterol turnover and phospholipid composition in an isoform dependent manner, whereas the expression levels of several lipid metabolism related genes also changed significantly. The results are interesting and informative. There are several comments about the current version of this manuscript.

  1. The authors should clarify some of the limitations of this study in the discussion part, like only male mice were used, lack of APOE heterzygous mice results, and etc.  

We concur that the point raised by the reviewer is indeed important: several studies have underlined a sex-depended association of APOE genotype and cognitive deficit. Indeed, women carrying the APOE4 genotype show a greater risk for developing AD pathology than men, particularly at younger ages (Farrer et al. JAMA 1997; PMID9343467; A recent study of Martinsen et al. 2020 [30] found lower levels of PUFA in female mice with increasing age, while no significant effect was observed in males. This comment has been  added to the introduction as follows (lines 75-79, page 2):

Another study focused on the analysis of fatty acids in the hippocampus and cortex of male and female APOE3 and APOE4 KI-mice. The analysis revealed an age related decrease in the levels of omega-3 fatty acid only in APOE4 females, but no alterations in males, underlining an interplay between sex, age and genotype”. 

Moreover, we have pointed out the limitation of this study concerning the absence of females in the discussion (page 12, line 437-439), as follows: ”One of the limitations of the study is that we focused only on male mice. Taking into account interactions between APOE genotype, lipid content and gender [30], it would be of interest to undertake further studies in females as well.”

 Concerning the mouse model used, mice are homozygous KI mice for the human APOE3 and APOE4 genes. The mouse Apoe gene was replaced by either the human APOE3 or APOE4 gene. 

  1. The authors detected cholesterol and phospholipid metabolism changes in APOE4 mice. How about the APOE level in mouse brain? Is there any correlation between lipid levels and APOE levels?

Previous studies showed lower levels of the APOE in the brain of APOE4 mice as compared to APOE3 (Mann et al. [36]). We have cited this reference in the discussion (lines 553-555, page 15).

  1. Using primary neurons, the authors measured Abeta, p-Tau and apoE levels, it's not new that APOE4 increases Abeta and p-Tau levels in neurons.  The novelty of the manuscript will be enhanced if the authors could measure the lipid changes in primary neurons using lipidomics analysis and evaluate how much neuronal lipid contributes to the lipid changes of the whole brain.

We did in fact measure cholesterol levels in APOE3 and APOE4 derived neurons but the results were not included in the first, submitted version of manuscript. Significantly lower levels of cholesterol were observed in APOE4 as compared to APOE3 neurons, a more pronounced difference as compared with whole brain content. We thus decided to further analyse lipid dysregulation in intact brains in order to get a clearer picture and to take into account cell diversity. In accordance with the reviewer’s suggestion, we have now integrated these results on cholesterol levels in APOE3 and APOE4 primary neurons on the revised version (page 6, line 276-278 and Figure 1).

Lipidomic analysis of primary neurons would be a valuable further step in the investigation of the link between APOE4 genotype, lipid metabolism and AD pathology and it would be warranted to design and effect a comprehensive future study incorporating this idea as suggested by the reviewer. 

Reviewer 3 Report

There are several shortcomings that must be addressed by the authors:

1)    While the content of unesterified cholesterol and its metabolites was measured beautifully, the content of esterified cholesterol was not. Given the known importance of lipid droplets in AD and the fact that cholesterol metabolism cannot fully be evaluated without taking into consideration the storage of cholesterol in the form of esterified cholesterol in lipid droplets this missing data is a major shortcoming of the study. It is also a shortcoming that triglycerides were not measured. There is some evidence in the literature that TG (as well as CE) are increased due to higher lipid droplet formation in AD brains. Can the authors explain why TG was not measured as a lipid class? Without the CE and TG measurement, it is not possible to state that this is a complete examination of lipid dysregulation overall, but instead of a subset of lipids. For that matter, DG, MG, lyso-PL, ceramides, and other lipid classes are also missing here. So again, it cannot be said that this is a complete examination of overall lipid metabolism, but instead an investigation into PC, PE and unesterified cholesterol. The authors need to change the language throughout the manuscript to reflect this. 

2)    The authors need to account for multiple comparisons in their statistical analysis throughout the manuscript but in particular in the sections evaluating the differences in fatty acid composition. It is likely that several of the statistically significant differences will not remain statistically significant after correction for multiple testing. And indeed, the differences do not appear to be large in magnitude, and do not always make sense. For example, it does not make sense that the increase in % of monounsaturated fatty acids is significantly different but there is no concomitant significant decrease in the % of saturated and/or polyunsaturated fatty acids. Since the total must add up to 100%, if one category is increased one or both of the others must be decreased.

3)    It would be important to examine whether the observed increases in Tau and pTau as well as Abeta in the neurons are related to the lipid composition alterations. Otherwise the first section of the paper detailing the differences in lipid composition and the second section of the paper detailing the differences in Abeta and Tau are somewhat disconnected from each other. 

4)    The discussion section needs to be toned down to more accurately portray the extent of the study. Since, as stated previously, the authors did not measure lipid composition comprehensively and since specific critical lipid classes were excluded, it is not accurate to state that this study examined the effects of APOE variants on “lipid metabolism” globally but rather on the subset of lipids that were measured in this study. Unlike what the authors state in lines 389-392, the current study is NOT in fact a comprehensive analysis of all lipids, but rather of just a subset of lipid classes, therefore it is not possible to make statements about overall changes in lipid metabolism.

5)    The discussion in lines 400-420 absolutely must point out the shortcoming of this study pointed out above, that esterified cholesterol and other lipid classes were not quantified, therefore a comprehensive picture of whether there was increased intracellular storage of cholesterol within lipid droplets as part of the observed alterations in cholesterol synthesis and degradation, as well as whether there were other alterations in lipid metabolism, is not possible. 

6)    It is critical to point out in the section in lines 428-441 that in addition to de novo synthesis of PC and PE, PE can also be converted to PC via PEMT. In fact, there is quite a robust literature showing that the specific composition of fatty acids within PC and PE can be indicative of diminished flux through this critical pathway for PC synthesis, which in fact, could explain some of the observations of differences in PC and PE FA composition. The authors should look into this literature and add some analyses of PEMT-specific FA species to determine whether changes in the PEMT pathway could be involved. 

7)    Accordingly, figure 8 should be updated to account for pathways that were not measured in addition to those that were

8)    There are other approaches for dealing with the problem discussed in lines 482-488. For example, it is possible to look at different ratios of products to precursors for each elongase and desaturase enzyme. There are several published papers using this approach that have demonstrated the ratios are better associated with observed changes in enzyme activity/expression than just the fatty acids themselves (e.g. papers looking at the desaturase index). The authors should look into this.

9)    All of the methods sections describing the lipid analysis need to be expanded. All of the sections need more information on how the GC and MS data were collected and processed. At minimum, the authors should cite a paper detailing the method and provide a couple of sentences briefly describing the procedure. In particular, information on how data are acquired and processed is needed, with detailed information on the data outputs (ion counts, normalized ion counts, etc), and whether/which internal standards were used. It is not adequate to simply state that “conventional methods” were used without even providing any citations.

10) Data processing and statistical analysis need to be separate sections detailing each method used for each separate type of lipid analysis.

11) How were the % of total content for phsopholipids calculated exactly? What was done with species that were present/quantified in only a subset of samples? Were they still included in the “total content” for all samples? As mentioned above, correction for multiple testing must be applied.

12) Lines 247-249: this is an example of language that needs to be more clear: the authors state that the “ratio of cholesterol/desmosterol as well as cholesterol/beta-cholestanol were significantly increased in APOE4 mice suggesting an imbalance between cholesterol and the other sterols”. Do the authors mean in relation to APOE3 mice? 

13) The section 3.4: “lipid dysregulations and gene expression in APOE genotypes” does not have a methods section anywhere in the manuscript and therefore the reader has no idea where the samples came from, how the RNAseq was completed, and how it is related to the current manuscript. If this is simply an examination of published data, it cannot be reported here as part of the “research” that was done in this study. It is OK to use information from these published data and include findings in the discussion section that are relevant to the findings from this study. However, simply reporting previously published data as part of the “results” of this paper is not an admissible practice.

14) In line 331: the authors state that there was a significant “overproduction” of the Abeta peptides in APOE4 neurons, but the only thing one can accurately state given the actual measurement performed is that there was a “higher production in APOE4 vs APOE3 neurons”

15) The section in lines 347-352 does not make sense. How does the fact that astrocytes are the main producers of ApoE in the brain have anything to do with intracellular concentrations of ApoE in neurons? Also there is no justification or evidence for the authors’ statement that ApoE “turnover probably increased”. All you can say given the data is that APOE4 was significantly lower in APOE4 neurons vs. APOE3 neurons. 

16) In lines 421-422: the authors need to be more careful about statements that cerebral cholesterol synthesis in APOE4 mice is impaired since the overall total cholesterol concentrations were not actually different. 

17) Animal use and safety protocols and approvals need to be included

18) Much of the discussion section is speculative and should be more carefully phrased as such

19) Lines 532-534, the discussion here about oleic acid load stimulating the activity of gamma secretase and augmenting Abeta production leaves out the fact that it is now clearly known that this is a lipid raft phenomenon and therefore the concentrations of cholesterol (and other lipid raft-specific lipids) in the plasma membranes of neurons are far more important determinants of this process.

Author Response

There are several shortcomings that must be addressed by the authors:

  • While the content of unesterified cholesterol and its metabolites was measured beautifully, the content of esterified cholesterol was not. Given the known importance of lipid droplets in AD and the fact that cholesterol metabolism cannot fully be evaluated without taking into consideration the storage of cholesterol in the form of esterified cholesterol in lipid droplets this missing data is a major shortcoming of the study. It is also a shortcoming that triglycerides were not measured. There is some evidence in the literature that TG (as well as CE) are increased due to higher lipid droplet formation in AD brains. Can the authors explain why TG was not measured as a lipid class? Without the CE and TG measurement, it is not possible to state that this is a complete examination of lipid dysregulation overall, but instead of a subset of lipids. For that matter, DG, MG, lyso-PL, ceramides, and other lipid classes are also missing here. So again, it cannot be said that this is a complete examination of overall lipid metabolism, but instead an investigation into PC, PE and unesterified cholesterol. The authors need to change the language throughout the manuscript to reflect this. 

We thank the reviewer for the remark. Indeed, we only focused on the main lipid classes present in the brain, the PC, PE and cholesterol. The reviewer is right to mention that other classes of lipids could have been studied, as TG, DGs, CE, sulfatides, free fatty acids, oxidized fatty acids – all known to be involved in AD. However it would have been difficult to incorporate all classes of lipid and we wished to focus on those of the greatest relevance and most directly implicated in the interrelationship between ApoE and neurotoxic proteins involved in AD. This is  now clearly mentioned in the manuscript – in the introduction and discussion sections - that we focused on the three major and most pertinent classes of membrane lipids, phoshatidylcholines, phosphatidylethanolamines and cholesterol as well as global fatty acids.

lines 82-89, page 2

In order to further identify new links between APOE genotype and lipid metabolism and to evaluate their co-joint impact on AD pathology, we first characterized the brain lipidomic profile of several classes of lipids in KI mice expressing human APOE3 or APOE4. We focused on three major and most relevant classes of membrane lipids: phoshatidylcholines, phosphatidylethanolamines and cholesterol. We also investigated the global composition in fatty acids, the essential components of most lipids, in order to embrace all lipid classes in the brain including the less abundant triglycerides and cholesteryl esters.”

lines 429-437, page 12 “In contrast to previous lipidomic studies that focused only on a particular class of lipids, either cholesterol or fatty acids, here we present a quantitative assessment of the most abundant lipids classes in the brain : cholesterol, phosphatidylcholines and phosphatidylethanolamines. We also established the global fatty acid profile to get an overview of all lipid classes.”

 The authors need to account for multiple comparisons in their statistical analysis throughout the manuscript but in particular in the sections evaluating the differences in fatty acid composition. It is likely that several of the statistically significant differences will not remain statistically significant after correction for multiple testing. And indeed, the differences do not appear to be large in magnitude, and do not always make sense. For example, it does not make sense that the increase in % of monounsaturated fatty acids is significantly different but there is no concomitant significant decrease in the % of saturated and/or polyunsaturated fatty acids. Since the total must add up to 100%, if one category is increased one or both of the others must be decreased.

In accordance with the reviewer’s suggestion, we have revised the analysis of the different lipid classes, using one way analysis of variance (ANOVA) followed by multiple comparison tests with Fisher correction. The Data processing section details now the new statistical tools used (page 6, lines 259-267).

As presumed by the reviewer, the use of multiple comparisons slightly modified the significance for certain lipids (FA 20:1 and FA 20:4, are no longer significantly different between the two genotypes; on the contrary, FA 18:1 became significant). We have modified the manuscript accordingly (page 8, line 3374-339), and adjusted Figure 3.

We agree that MUFA+PUFA+Saturated FA represent 100% of total FA. We found a significant increase of MUFA in APOE4 brains compared to APOE3. Accordingly, the levels of PUFA and saturated FA were both decreased (from 20.35% and  54.06% respectively for APOE3 to 20% and 53.75% respectively for APOE4), although not significantly, but the balance between MUFA and PUFA+Satutared FA was indeed maintained.

  • It would be important to examine whether the observed increases in Tau and pTau as well as Abeta in the neurons are related to the lipid composition alterations. Otherwise the first section of the paper detailing the differences in lipid composition and the second section of the paper detailing the differences in Abeta and Tau are somewhat disconnected from each other. 

We proposed several hypotheses on the potential link between lipid dysregulations and AD pathology in the paragraph “Interrelationship between lipid dysregulation and AD pathological markers” in the discussion section. Still, despite the present novel information, much remains to be  done in order to fully understand the interplay between lipid homeostasis, APOE genotype and AD pathology.

As mentioned in the conclusions (593-597) “Future studies should examine more closely the precise molecular mechanisms interlinking lipid dysregulations with APOE4 genotype, in particular, the altered generation and processing of cholesterol and phospholipids. Pathways linking lipid dys-homeostasis to the anomalous generation of neurotoxic peptides and other pivotal cellular substrates of AD also merit further characterisation.”

4)    The discussion section needs to be toned down to more accurately portray the extent of the study. Since, as stated previously, the authors did not measure lipid composition comprehensively and since specific critical lipid classes were excluded, it is not accurate to state that this study examined the effects of APOE variants on “lipid metabolism” globally but rather on the subset of lipids that were measured in this study. Unlike what the authors state in lines 389-392, the current study is NOT in fact a comprehensive analysis of all lipids, but rather of just a subset of lipid classes, therefore it is not possible to make statements about overall changes in lipid metabolism.

We have modified the discussion accordingly and now underline the fact that we only analysed the most abundant classes of lipids in the brain as previously detailed (remark 1).

5)    The discussion in lines 400-420 absolutely must point out the shortcoming of this study pointed out above, that esterified cholesterol and other lipid classes were not quantified, therefore a comprehensive picture of whether there was increased intracellular storage of cholesterol within lipid droplets as part of the observed alterations in cholesterol synthesis and degradation, as well as whether there were other alterations in lipid metabolism, is not possible. 

We have now clarified that not all classes of lipids were analysed (see remark 1). We did not originally explain why we did not analyse TG and CE in order to avoid overloading the manuscript with information and distracting from the core results and their interpretation. In fact, the reason that we did not analyse TG or Cholesteryl esters is that, according to previous lipidomic analysis in mice and human brain tissue, TG or Cholesteryl esters represent  less than 2% expressed as mean mol% of all lipid species (Chan et al. JBC 2012; PMID 22134919). Moreover, the mice used for the study were 12 months old. As lipid droplets are age dependent, their content in the brain of the mice we used will have been very low, according to Shimabukuro et al. Sci. Rep. 2016 (PMID 27029648).

 To reiterate, we believe that mentioning all the lipid classes not actually analysed, and their putative role in AD pathology would have over-encumbered the discussion and compromised its clarity and fluidity.  

6)    It is critical to point out in the section in lines 428-441 that in addition to de novo synthesis of PC and PE, PE can also be converted to PC via PEMT. In fact, there is quite a robust literature showing that the specific composition of fatty acids within PC and PE can be indicative of diminished flux through this critical pathway for PC synthesis, which in fact, could explain some of the observations of differences in PC and PE FA composition. The authors should look into this literature and add some analyses of PEMT-specific FA species to determine whether changes in the PEMT pathway could be involved. 

We included this pathway in the manuscript (lines 478-479). PEMT gene expression was not modulated in the RNAseq data, suggesting that this pathway was probably not impacted by the APOE genotype (line 483).

7)    Accordingly, figure 8 should be updated to account for pathways that were not measured in addition to those that were

As suggested, we have  added this pathway to Figure 8.

8)    There are other approaches for dealing with the problem discussed in lines 482-488. For example, it is possible to look at different ratios of products to precursors for each elongase and desaturase enzyme. There are several published papers using this approach that have demonstrated the ratios are better associated with observed changes in enzyme activity/expression than just the fatty acids themselves (e.g. papers looking at the desaturase index). The authors should look into this.

These points are indeed of not inconsiderable importance yet they are actually out of the scope of the present study. Their further investigation would enhance out understanding the link between lipid metabolism and the progression of the disease, and this substantial enterprise would be fully justified as part of a future set of studies.

9)    All of the methods sections describing the lipid analysis need to be expanded. All of the sections need more information on how the GC and MS data were collected and processed. At minimum, the authors should cite a paper detailing the method and provide a couple of sentences briefly describing the procedure. In particular, information on how data are acquired and processed is needed, with detailed information on the data outputs (ion counts, normalized ion counts, etc), and whether/which internal standards were used. It is not adequate to simply state that “conventional methods” were used without even providing any citations.

In line with a request from the editor, we have improved the description of mass spectrometry methodology, including parameters of the analysis, software used for peak detection and data analysis as follows:

Lines 133-142: “GC-MS/MS was equipped with a SolGel-1ms fused silica capillary column (Trajan, SGE; 60m×0.25mm). The carrier gas was helium (1,2ml/min). Temperature of the split/splitless injector was set at 280°C. The oven temperature program was as follows: 55°C for 4 min, followed by 40°C/min up to 250°C, and 20°C/min up to 310°C (for 25 min). The samples were injected in a splitless mode. Temperatures of the mass spectrometer transfer line and the source were set at 250°C and 20 °C respectively. Nitrogen was used as collision gas (1.5ml/min). The electron ionization energy was 70eV. Cholesterol was detected using MRM (Multiple reaction monitoring) mode. Peak detection, integration and quantitative analysis were executed using MassHunter software.”

Lines 150-167: “High performance liquid chromatography was performed using a Shimadzu Nexera LC-30 equipped with an autosampler, a binary pump and a column oven. The analytical column was a WATERS CORTECS HILIC (1.6μm, 3.0×150mm). Separation was carried out at 40°C using a linear gradient. Mobile phase A consisted of 50% 10mM ammonium formate and 50% acetonitrile. Mobile phase B was 5% 10mM ammonium formate and 95% acetonitrile. The mobile phase gradient was delivered as follows: 100% B from 0 to 2 min, 0% B at 10 min then return to the initial conditions. The flow rate was 0.7 ml/min and 5μl sample volume was injected. The HPLC system was coupled on-line to a QTRAP 4500 (Sciex) equipped with electrospray ionization source (ESI). Source parameters were as follows: source temperature was set at 350 °C, curtain gas at 40, gas 1 at 20 and gas 2 at 35, using nitrogen. Analyses were achieved in positive mode, based on precursor ion scan (m/z 184) for PC and neutral loss (m/z 141) for PE. Spray voltage was at 4500 V. Nitrogen is used as collision gas. Finally, peak detection, integration and quantitative analysis were performed using Analyst and Lipidview softwares (Sciex).”

 Lines 176-183: “GC was performed on an HP 6890 (Agilent Technologies) instrument equipped with a fused silica capillary BPX70 column (Trajan, SGE; 60mX0.25mm). The carrier gas was hydrogen (1ml/min). Temperatures of the flame ionization detector and the split/splitless injector were set at 250°C and 230°C, respectively. The oven temperature program was as follows: 50°C for 2 min, followed by 20°C/min up to 140°C, and 2°C/min up to 240°C (for 5 min). The samples were injected in a splitless mode. Peak detection, integration and quantitative analysis were executed using MassHunter software.”

This section has accordingly been rendered more clear and comprehensive.  

The use of internal standards was mentioned in the paragraph describing the extraction in the Mat Met chapter. (50µg or PC di-17:0, PE di-17:0 and, 5µg of cholesterol-2,3,4-13C) (see lines 103-107, page 3): “Total brain homogenates (approx. 130mg) were extracted according to Folch procedure (FOLCH et al., 1957): 2ml of cold methanol-BHT (butyl-hydroxytoluene) spiked with several standards (50µg or PC di-17:0, PE di-17:0 and, 5µg of cholesterol-2,3,4-13C) were added to the homogenate and stirred for 10 min, at 4°C.”

10) Data processing and statistical analysis need to be separate sections detailing each method used for each separate type of lipid analysis.

We thank the reviewer for the remark. We have as requested, now detailed for each lipid class, the analysis and statistics performed (250-267, page 6): Cholesterol and derivatives content in the brain were calculated per g of tissue with respect to 2,3,4-13C cholesterol used as (IS). Average and standard error were estimated over 7 samples for APOE3 genotype and 8 samples for APOE4 genotype (one APOE3 sample was formerly used for the preliminary set-up). For the rest of the analysis, 8 samples per phenotype were used. The proportion of PC and PE was calculated as molar percentage of the total phospholipid content (PC+PE). Only species present in at least 14 of the 16 samples were analysed. The relative amount of the excluded species was inferior to 1% of global phospholipid amount. Relative contents of individual species within a class of phospholipids were estimated as molar percentage of total lipid of the respective class. Only the species with a relative abundance greater than 1% for at least one of the samples were considered. The proportion of different fatty acids was also expressed as molar percentage of total fatty acids detected. Details on lipidomic analysis are available in supplementary material (Table S2 and S3). Statistical analysis of lipids was performed with Origin Pro software (Origin Lab), using multiple paired student test and one way analysis of variance (ANOVA) followed by multiple comparison tests with Ficher’s correction. Statistical significance was defined as: * for p < 0.05, ** for p<0.005 and ***: p <0.001.”

This has indeed improved the clarity of this section.

11) How were the % of total content for phosopholipids calculated exactly? What was done with species that were present/quantified in only a subset of samples? Were they still included in the “total content” for all samples? As mentioned above, correction for multiple testing must be applied.

As mentioned above, we have now detailed in the manuscript that total phospholipid content represents the sum of PC+PE and that only those species of phospholipid present in at least 14 of 16 samples analysed were considered for further statistical analysis. The relative amount of the excluded species was below 1% of the global amount of phospholipids and should not have affected the outcome of analyses (page 6, lines 256-258).

Multiple comparisons were performed for the analysis of PC and PE, as suggested by the reviewer and the modifications in the statistical significance were implemented in the manuscript (see line 312-318, page 7). According to the new analysis, the level of PC 18:0/18:1 and PE 18:0/18:1 was not significantly different between the two genotypes; on the contrary, PE16:0/22:6 and PE 18:0/22:6 became significant. These changes were minor and did not impact the global message of the study. Figure 2 corresponding to the analysis of PC and PE species was also corrected.

12) Lines 247-249: this is an example of language that needs to be more clear: the authors state that the “ratio of cholesterol/desmosterol as well as cholesterol/beta-cholestanol were significantly increased in APOE4 mice suggesting an imbalance between cholesterol and the other sterols”. Do the authors mean in relation to APOE3 mice? 

We thank the reviewer for the remark. Indeed, we meant that “the ratio cholesterol/desmosterol as well as cholesterol/β-cholestanol were significantly increased in APOE4 mice compared to APOE3 mice”. We have corrected this inaccuracy in the revised manuscript (line 289-291, page 6).

13) The section 3.4: “lipid dysregulations and gene expression in APOE genotypes” does not have a methods section anywhere in the manuscript and therefore the reader has no idea where the samples came from, how the RNAseq was completed, and how it is related to the current manuscript. If this is simply an examination of published data, it cannot be reported here as part of the “research” that was done in this study. It is OK to use information from these published data and include findings in the discussion section that are relevant to the findings from this study. However, simply reporting previously published data as part of the “results” of this paper is not an admissible practice.

As stated in the introduction (page 2, lines 89-91): “We further examined the link between lipid dysregulations and gene expression based on a recently published RNAseq data set [31]”

 14) In line 331: the authors state that there was a significant “overproduction” of the Abeta peptides in APOE4 neurons, but the only thing one can accurately state given the actual measurement performed is that there was a “higher production in APOE4 vs APOE3 neurons”

We have corrected this inaccuracy (page 10, line 388-390): “We observed a significant enrichment of the three main Aβ fragments: 38, 40 and 42 in APOE4- as compared to APOE3-derived neurons that was not due to an overexpression of the Amyloid Precursor Protein (APP).”

15) The section in lines 347-352 does not make sense. How does the fact that astrocytes are the main producers of ApoE in the brain have anything to do with intracellular concentrations of ApoE in neurons? Also there is no justification or evidence for the authors’ statement that ApoE “turnover probably increased”. All you can say given the data is that APOE4 was significantly lower in APOE4 neurons vs. APOE3 neurons. 

The comment was indeed ambiguous in the previous version. We actually meant that “APOE was significantly reduced in APOE4-derived neurons” (line 392-393 page 11). This has now been clarified

16) In lines 421-422: the authors need to be more careful about statements that cerebral cholesterol synthesis in APOE4 mice is impaired since the overall total cholesterol concentrations were not actually different. 

We took into account the reviewer’s remark and revised the comment as follows (page 13, line 467-468): The present study suggests that cerebral cholesterol turnover is impaired in APOE4-KI mice”.

We also revised (removed) the statements concerning cholesterol transport, a mechanism indeed that could not be explored in the study yet warrants further work

17) Animal use and safety protocols and approvals need to be included

In this study, we collected the brains post-mortem, a procedure that does not require specific authorization according to official procedures where the work was performed

18) Much of the discussion section is speculative and should be more carefully phrased as such

19) Lines 532-534, the discussion here about oleic acid load stimulating the activity of gamma secretase and augmenting Abeta production leaves out the fact that it is now clearly known that this is a lipid raft phenomenon and therefore the concentrations of cholesterol (and other lipid raft-specific lipids) in the plasma membranes of neurons are far more important determinants of this process.

We have tried to formulate the discussion in an objective and factual manner. Regarding this specific point,  besides a decrease in membrane fluidity due to lower levels of PE-DHA that stimulate the interaction between APP and BACE1, an additional effect comprised by the increase of OA could stimulate gamma-Secretase activity. But we fully agree that lipid raft composition (mainly cholesterol) plays a crucial role in APP processing, as shown in our previous studies (Marquer et al. 2014, PMID 25524049; Marquer et al. 2011, [82]).

Round 2

Reviewer 2 Report

The authors have successfully addressed all of my concerns. The manuscript is qualified to be accepted for further publication.